# Clinical and Molecular Profiling of Colorectal Cancer: A Comprehensive Cohort Study of BRAF-Mutated Cases from a Tertiary Centre

**DOI:** 10.3390/curroncol32090507

**Published:** 2025-09-12

**Authors:** Julia Freckelton, Justin Mencel, Iris Levink, Sheela Rao, Charlotte Fribbens, Paula Proszek, Damian Brooks, Xin Liu, David Cunningham, Ian Chau, Naureen Starling

**Affiliations:** 1Royal Marsden Hospital, London SW3 6JJ, UKi.levink@erasmusmc.nl (I.L.); ian.chau@rmh.nhs.uk (I.C.);; 2Institute of Cancer Research, London SW7 3RP, UK

**Keywords:** BRAF, colorectal cancer, molecular

## Abstract

Colorectal cancer is a common disease, and a specific genetic change called a BRAF mutation is often used by doctors to guide treatment decisions. However, not all BRAF mutations are the same, and patients with these mutations can experience very different outcomes. This study looked back at 406 patients with BRAF-mutated colorectal cancer to better understand how different types of BRAF mutations and other genetic changes affect how the disease behaves and responds to treatment. The research showed that patients with one type of BRAF mutation (called V600) tended to be older, have tumours on the right side of the bowel, and were more likely to see the cancer return quickly after treatment. On the other hand, patients with a different type of BRAF mutation (non-V600) had better survival and more often had other gene changes as well. One key finding was that a separate mutation, called RNF43, was linked to better response to a common targeted treatment combination. These results suggest that testing for the specific type of BRAF mutation and other related gene changes can help doctors provide more accurate, personalised treatment plans for people with colorectal cancer.

## 1. Introduction

Colorectal cancer is the third most common cancer globally and responsible for the second highest number of cancer deaths [1]. Molecular assessment of colorectal cancer is now standard practice and is used to more accurately predict the behaviour of a cancer and to personalise selection of efficacious treatments. BRAF mutations are one of the most common oncogenic mutations, occurring in approximately 10% of colorectal cancers [2]. 

BRAF signals within the mitogen-activated protein kinase (MAPK) pathway, which governs cell growth and division, in response to extracellular stimulation. Oncogenic mutations in the BRAF can result in aggressive malignant behaviour via RAS-independent activation of MEK and ERK, resulting in uncontrolled cellular proliferation and survival [3]. 

The most common BRAF mutation is the V600E substitution, accounting for about 80% of all BRAF mutations in colorectal cancer [4]. ^V600E^BRAF mutations are most common in right-sided, proximal primary tumours. They are more likely to be chemotherapy refractory and pertain to a poor prognosis in metastatic colorectal cancer [5]. This has been demonstrated across various clinical contexts, including treatment with doublet chemotherapy regimens, treatment with triplet regimens [6] and after metastasectomy [7]. Unlike other cancers with a high prevalence of ^V600E^BRAF mutations, colorectal cancers with ^V600E^BRAF mutations are resistant to BRAF inhibitor monotherapy treatment. This is due to compensatory upregulation of downstream epidermal growth factor (EGFR) [8]. For this reason, dual blockade of BRAF and EGFR, using encorafenib and cetuximab, is an effective second-line systemic treatment for some patients harbouring a ^V600E^BRAF mutation, with an overall response rate (ORR) of 20% demonstrated in the BEACON trial [9]. There is also emerging evidence for benefit in the first line from the BREAKWATER study [10].

Several other BRAF mutations are linked to malignancies and are generally associated with a less aggressive disease phenotype [11]. BRAF mutations can be classified based on their impact on kinase activity. Class I mutations encompass all mutations on the V600 codon and typically result in strong RAS-independent kinase activity. Class II mutations are non-V600 mutations but also demonstrate RAS-independent kinase activity, but this tends to be weaker than that of Class I. Class III mutations retain RAS-dependent signalling of the MAPK pathway. 

Microsatellite instability high (MSI-H) and mismatch repair deficient status are more common in BRAF-mutated colorectal cancer. This is explained by an association with the CPG island methylator phenotype, which results in DNA promoter region hypermethylation. Promoter hypermethylation leads to silencing of gene expression (function to repair DNA mismatches), resulting in a microsatellite unstable tumour [12]. The efficacy of immunotherapy in these immune phenotype colorectal cancers is well established [13].

Mutations in Ring Finger Protein 43 (RNF43), a tumour suppressor gene, are more common in patients with ^V600E^BRAF mutations [14]. RNF43 regulates Wnt signalling, which is a pathway that has a well-described role in cancer pathogenesis [15]. Mutations in RNF43 are now understood to be present in multiple cancer types; however, they were first identified in colorectal cancer. They occur at a frequency of approximately 30% of ^V600E^BRAF colorectal cancers [16]. RNF43 works to modulate the Wnt pathway, so loss-of-function mutations in RNF43 can result in upregulation of this pathway [17].

There is significant heterogeneity in cancer behaviour and clinical outcomes amongst patients with BRAF-mutated colorectal cancer. There is emerging work providing an explanation for some of the variability in this patient group; however, it remains incompletely understood. This real-world study aims to describe the molecular profile, clinical phenotype, and patient outcomes of BRAF-mutated colorectal cancer treated at a single tertiary referral cancer centre in the United Kingdom. 

## 2. Materials and Methods

This retrospective cohort study included sequential adult patients with BRAF-mutated colorectal cancer, who received treatment between 2014 and 2022, at a single, tertiary referral cancer centre in the United Kingdom. All sequential BRAF mutant colorectal tumours were identified from the hospital’s molecular testing database. Patients were excluded if they did not have 12 months of follow-up information available. 

Response assessment was based on the outcomes reported in pre-existing clinical imaging reports from routine care. Disease control rate (DCR) was defined as the presence of stable disease or tumour response as documented at the first restaging imaging.

Patient demographic, health, and cancer treatment and outcome information was identified from the electronic health record. Information regarding tumour mutational subtype was identified from next-generation sequencing (NGS) reports. The NGS analysis was performed using DNA capture and targeted sequencing using the RMH200 solid panel. the detection of variants is dependent on the percentage of tumour infiltration, DNA input concentration and DNA quality. Confidence and minimum read depth are adjusted for tumour content. Minimum variant allele depth is 10×. Variant Allele Frequency (VAF) threshold at 5% can be detected with >98% sensitivity (95% CI) and >96% specificity (95% CI). Small indels are detected with sensitivity > 89% (95% CI) and accuracy > 73% (95% CI) at >5% VAF. This panel is capable of detecting gene amplifications (CNV ratio > 2) and deletions (CNV ratio < 0.5). 

BRAF mutations were classified as ^V600^BRAF-mutated or ^non-V600^BRAF-mutated. Further classification, into classes 1, 2 and 3, was performed using genomic data published by Chen et al. [18]. MMR status was determined using immunohistochemistry. 

IBM SPSS Statistics (Version 29.0, Chicago, IL, USA) software was used for all statistical analysis. Continuous data is presented as median (interquartile range) and tested with a Mann–Whitney U test to determine the significance of the relationship between categories, given its non-parametric nature. Survival was estimated using the Kaplan–Meier method, and a log-rank test was performed to assess the difference between survival curves. A *p*-value of less than 0.05 is considered significant in all analyses.

This project was approved by the Royal Marsden Hospital Trust institutional review board as a service evaluation (SE972).

## 3. Results

443 sequential patients with a diagnosis of BRAF-mutated colorectal cancer were identified between 2014 and 2022. 37 patients were excluded from the analyses due to loss to follow-up within 12 months of testing. Of the 406 patients included in the study, 253 had localised disease, and 153 had metastatic disease (Figure 1).

### 3.1. Part A: Localised Disease

Of the 253 patients with localised colorectal cancer, 228 had a ^V600^BRAF mutation, and 25 had a ^non-V600^BRAF mutation. Of the atypical mutations, 3 were class 2, 17 were class 3 and 5 could not be classified. Mismatch repair deficiency and older age were significantly more common in patients whose tumours also harboured a ^V600^BRAF mutation, compared to those with ^non-V600^BRAF mutations (*p* < 0.01, *p* = 0.04, respectively). Most patients had their primary cancer resected, but fewer received adjuvant chemotherapy: 45% and 60% of the patients with V600 and ^non-V600^BRAF mutations, respectively (Table 1). The rate of relapse was 40%, the same in both groups; however, the time to relapse was significantly longer in the ^non-V600^BRAF group (*p* = 0.006). 

### 3.2. Part B: Metastatic Disease

There were 153 patients with de novo metastatic colorectal cancer, and of these, 137 had a ^V600^BRAF mutation and 16 had a ^non-V600^BRAF mutation. In this cohort, 20% of patients with a ^V600^BRAF mutation had a mismatch repair deficiency, and all patients with a ^non-V600^BRAF mutation were mismatch repair proficient. KRAS mutations were significantly more common in patients with ^non-V600^BRAF mutations than in those with ^V600^BRAF mutations (27% vs. 1%, *p* < 0.01), as were NRAS mutations (14% vs. 3%, *p* = 0.04) and PIK3CA mutations (33% vs. 8%, *p* = 0.02) (Table 2). 

The median survival in patients with metastatic ^V600^BRAF-mutated colorectal cancer was 14 months, and in patients with metastatic ^non-V600^BRAF-mutated colorectal cancer, median overall survival was 34 months (Figure 2). 

### 3.3. Part C: RNF43 Mutation

Of the 158 patients tested with the NGS panel, which included both the BRAF and RNF43 genes, 47 patients (30%) were found to harbour the RNF43 mutation. The RNF43 mutation was associated with older age compared to RNF43 wild type (median 76 years vs. 64 years, *p* = 0.04) and higher frequency of right-side disease (80% vs. 58%, *p* = 0.01) (Table 3).

Patients with an RNF43-mutated tumour were more likely to have a ^V600^BRAF mutation compared to patients who were RNF43 wild type (98% vs. 79%, *p* = 0.02). Additionally, they were more likely to have a mismatch repair deficiency (63% vs. 27%, *p* = 0.01). TP53 mutations were significantly more common in patients who were RNF43 wild type compared to those were RNF43-mutated (60% vs. 25%, *p* = 0.01). 

Thirty-five patients received combined EGFR and BRAF inhibition treatment. Of these, a best response of disease control was achieved in a significantly higher proportion of patients with an RNF43 mutation, compared to those who were wild type (100% vs. 54%, *p* = 0.02) (Figure 3A). There was no difference in rate of response to immunotherapy between the RNF43-mutated and wild-type groups (*p* = 0.91) (Figure 3B). 

There was no significant difference in overall survival between the cohort of patients with metastatic, RNF43 wild-type colorectal cancer compared to those with metastatic RNF43-mutated colorectal cancer (Figure 4). 

## 4. Discussion

This study provides a well-described, real-world cohort of 406 patients with BRAF-mutated colorectal cancer. Our data is largely consistent with the existing literature and adds strength to previously observed characteristics of ^V600^BRAF and ^non-V600^BRAF-mutated colorectal cancer. ^Non-V600^BRAF mutations did occur at a lower frequency, compared to ^V600^BRAF mutations, than in previously reported cohorts [19]. This might be explained by the time period that this retrospective study covered, when BRAF testing was not reflexive. Our data is likely to represent patients with a more aggressive disease phenotype (associated with the ^V600^BRAF mutation) that later relapsed, because this would trigger clinicians to arrange molecular testing. For this reason, some patients with ^non-V600^BRAF may not be represented in the cohort. The time to relapse with ^non-V600^BRAF was significantly longer than with ^V600^BRAF, which is consistent with a less aggressive disease phenotype. Other key findings were that mismatch repair deficiency is rare and significantly less common in colorectal cancers with ^non-V600^BRAF mutations than in those with ^V600^BRAF mutations. ^Non-V600^BRAF mutations are associated with a higher frequency of KRAS, NRAS and PIK3CA mutations. Finally, coexisting RNF43 mutations were associated with a significantly higher disease control rate than patients with RNF43 wild-type status when treated with combined BRAF/EGFR therapy. 

We report a mismatch repair deficiency frequency of 20% in patients with metastatic ^V600^BRAF-mutated CRC, but this was significantly less common in patients with ^non-V600^BRAF mutations. This is concordant with other published data sets [20]. The molecular profile described in our metastatic cohort of patients is also consistent, demonstrating a higher prevalence of KRAS mutations in patients with ^non-V600^BRAF mutations. 

In our cohort, 30% of patients tested had a co-existing RNF43 mutation, which is consistent with previously reported prevalences in patients with BRAF-mutated colorectal cancers [18]. RNF43 mutation was associated with older age, earlier-stage disease at presentation, right-side disease and ^V600^BRAF mutations. The increased frequency of TP53 mutations in patients with RNF43 wild-type tumours may be partly explained by the higher prevalence of class 3 BRAF mutations in this group. Class 3 BRAF mutations have previously been shown to be associated with high rates of TP53 mutations [18].

We demonstrated that RNF43 mutation was associated with a significantly higher disease control rate from combination BRAF and EGFR inhibition therapy than in those who were RNF43 wild type. This is consistent with the cohort described by Elez et al. [16], which demonstrated improved progression-free survival on combined BRAF and EGFR inhibition therapy in patients with RNF43-mutated tumours compared to those with wild-type tumours. It adds further weight to the hypothesis that RNF43 mutation could be used as a positive biomarker to predict response to combined BRAF/EGFR inhibition therapy. There was no difference in median progression-free survival in our group; however, this study was limited by a small number of patients. It is important to note, however, that in the subgroup analysis of the Beacon trial, the subgroup analysis only showed benefit from encorafenib/cetuximab over chemotherapy in the RNF43 wild-type group and not the RNF43-mutated groups [21].

Due to the rare nature of ^non-V600^BRAF mutations and of co-existing RNF43 mutations and BRAF mutations, much of the analysis was limited by small sample size. This also limited the ability to perform muti-variable analyses to control for confounding factors. A change in the availability of treatments for BRAF-mutated CRC over the past decade, particularly the introduction of dual BRAF and EGFR inhibition for ^V600^BRAF-mutated CRC and checkpoint inhibition for dMMR CRC into standard of care practice, limits the interpretation of the older cohort. The localised disease cohort also needs to be interpreted with caution. The transcriptional subtypes of ^V600E^BRAF-mutated colorectal cancer (BM1 and BM2) were not tested for; however, they could provide further predictive information for this group of patients [22].

## 5. Conclusions

There is significant heterogeneity in presentation, behaviour and outcomes in patients with BRAF-mutated colorectal cancer. Our real-world data confirms some of these differences can be explained by BRAF mutational class and the presence of concurrent RNF43 mutations. Further characterisation of these patient groups could assist with more accurately predicting response to therapies and personalising treatment approaches. Our data suggests a predictive role for RNF43 mutation in response to encorafenib and cetuximab and when viewed in the context of other available data, suggests the utility of routine testing of RNF43 in patients with BRAF-mutated colorectal cancer, if external validation is supportive.

## Figures and Tables

**Figure 1 curroncol-32-00507-f001:**
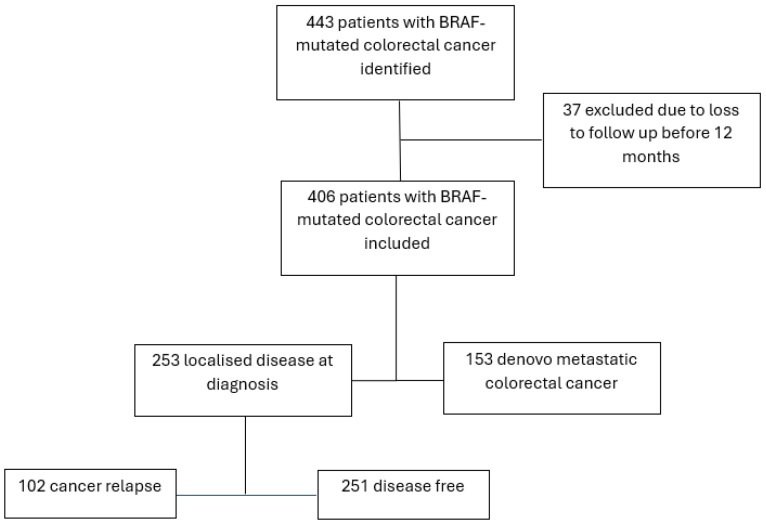
Included patients.

**Figure 2 curroncol-32-00507-f002:**
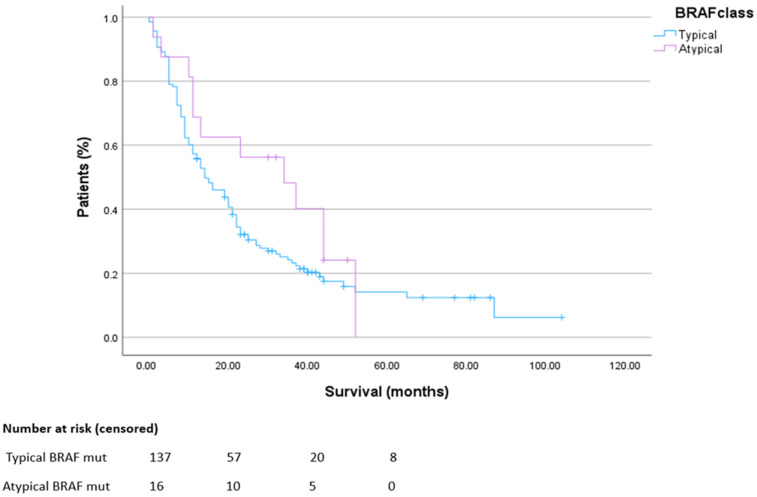
Overall survival in metastatic disease. HR 0.69 CI 0.41–1.15. *p* = 0.12.

**Figure 3 curroncol-32-00507-f003:**
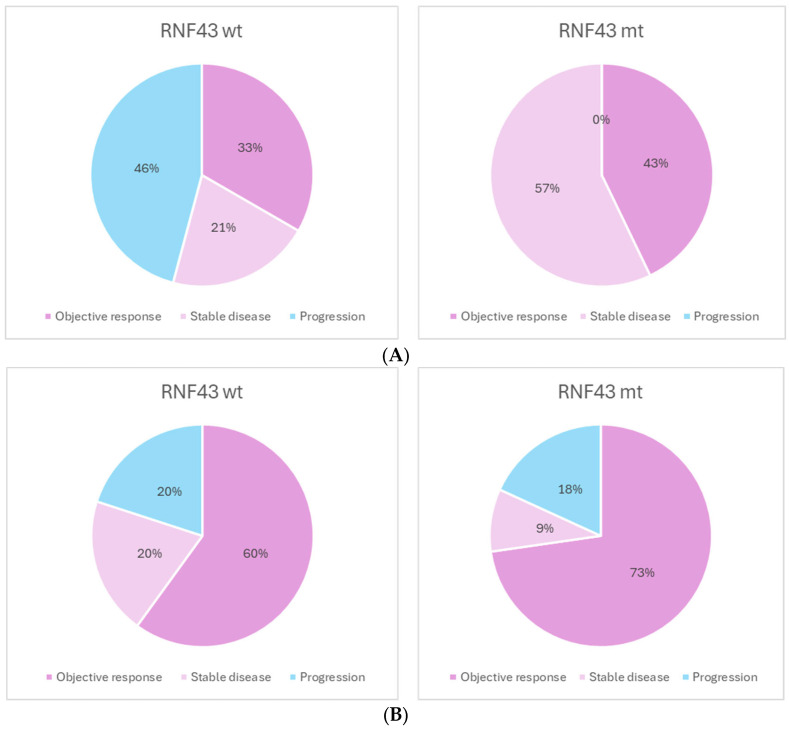
(**A**). Best response to BRAF/EGFR inhibition. N = 35. DCR: 54% vs. 100% *p* = 0.02. (**B**). Best response to checkpoint inhibition. N = 21. DCR: 80% vs. 82% *p* = 0.9.

**Figure 4 curroncol-32-00507-f004:**
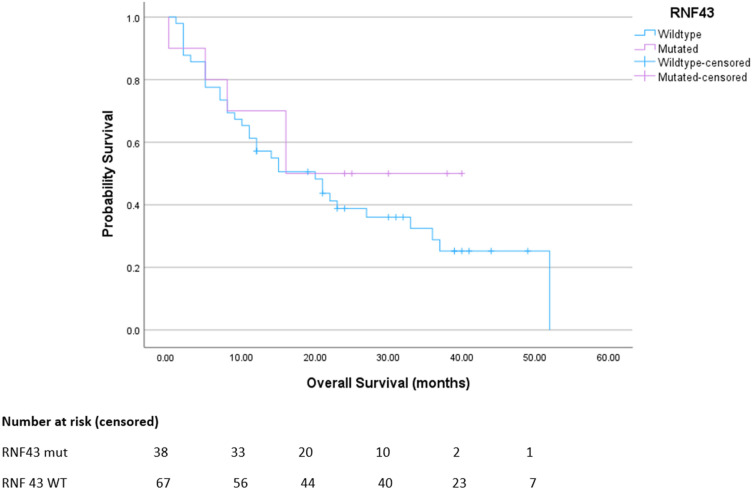
Overall survival for advanced BRAFmt CRC. Median survival for RNF43 wt was 20 months; mut was not reached. *p* = 0.39. HR 0.87 (95% CI 0.46–1.64).

**Table 1 curroncol-32-00507-t001:** Patient, tumour and treatment characteristics in localised disease.

	^V600^BRAFmt (N = 228)	^Non-V600^BRAFmt (N = 25)	*p*-Value
**Demographic information**			
Age, years median (IQR)	73 (63–79)	63 (54–71)	0.04
Male, n (%)	136 (60)	12 (48)	0.3
**Tumour characteristics**			
Stage at diagnosis, n (%) I II III	12 (5) 60 (26) 156 (68)	3 (12) 8 (32) 14 (56)	
Right-side primary	166 (73)	10 (40)	<0.01
BRAF mutation Class 1 Class 2 Class 3 Unclassified	228	3 17 5	
MMRd, n (%) MLH1 + PMS2 loss PMS2 loss MSH2 + MSH 6 loss MLH1 + PMS 2 + MSH 6 loss	128 (56) 117 7 2 2	2 (8) 0 1 0 1	<0.01
**Treatment**			
Resection of primary, n (%)	213 (93)	24 (96)	0.6
Adjuvant systemic treatment	100 (44)	15 (60)	0.1
Adjuvant treatment received Doublet Single agent	57 43	11 4	
**Outcome**			
Relapse, n (%)	92 (40)	10 (40)	0.97
Time to relapse, median months (IQR)	8 (3–13)	25 (7–58)	0.006

Mismatch repair deficient (MMRd), interquartile range (IQR), BRAF mutant (BRAFmt).

**Table 2 curroncol-32-00507-t002:** Patient, tumour and treatment characteristics in metastatic disease.

	^V600^BRAF mut (N = 137)	^Non-V600^BRAF mut (N = 16)	*p*-Value
**Demographic information**			
Age (years), median (IQR)	64 (43–75)	64 (56–77)	0.5
Male, n (%)	66 (48)	5 (43)	0.2
**Tumour characteristics**			
Right-side primary	86 (63)	9 (56)	0.3
BRAF mutation Class 1 Class 2 Class 3 Non-V600, class unknown	137	2 11 3	
MMR deficiency, n (%) MLH1 + PMS2 loss PMS2 loss MSH2 + MSH 6 loss	28 (20) 24 (89) 2 (7) 1 (4)	0	0.01
KRAS mutation, n (%) N = 149	1 (1)	4 (27)	<0.01
NRAS mutation, n (%) N = 145	4 (3)	2 (14)	0.04
PIK3CA mutation, n (%) N = 107	8 (8)	4 (33)	0.02
TP53 mutation, n (%) N = 101	54 (60)	7 (64)	0.8
RNF43 mutation, n (%) N = 58	10 (18)	0 (0)	0.6
**Treatment**			
Number of SACT regimens received, n (%) 0 1 2 3 4 >4	19 (14) 42 (31) 30 (22) 29 (21) 8 (6) 9 (7)	1 (6) 5 (31) 2 (13) 6 (38) 1 (6) 1 (6)	
Treatments received: Fluoropyrimidine/oxaliplatin Fluoropyrimidine/irinotecan FOLFOXIRI PD1/PDL1 blockade BRAF/EGFR inhibition Trifluridine/tipiracil	81 65 17 13 39 11	13 12 0 0 0 4	

IQR = interquartile range. MMR = mismatch repair. SACT = systemic anticancer treatment. FOLFOXIRI = fluoropyrimidine, oxaliplatin, and irinotecan. Percentages are percentages of the total number with mutation. In cases where data was missing, the reduced number is presented under the characteristic.

**Table 3 curroncol-32-00507-t003:** RNF43 mutation in BRAF-mutated colorectal cancer.

	RNF43 Wild Type (n = 111)	RNF43 Mutation (n = 47)	*p*-Value
**Demographics**			
Male, n (%)	42 (38)	22 (47)	0.7
Age (years), median (IQR)	64 (53–76)	76 (64–80)	0.04
**Cancer characteristics**			
Stage at diagnosis n (%) I II III IV	6 (5) 14 (13) 42 (38) 49 (44)	3 (7) 8 (17) 26 (57) 9 (20)	0.01
Primary location Right Left Both	64 (58) 47 (42) 0	37 (80) 8 (17) 2 (3)	0.01
MMR deficiency, n (%)	30 (27)	29 (63)	0.01
BRAF mutation, n (%): Class 1 Class 2 Class 3 Non-V600, class unknown	88 (79) 2 (2) 14 (13) 7 (6)	46 (98) 1 (2) 0 (0) 0 (0)	0.02
KRAS mutation, n (%) N = 154	6 (6)	2 (4)	0.99
NRAS mutation, n (%) N = 154	3 (3)	2 (4)	0.57
PIK3CA mutation, n (%) N = 64	11 (20)	3 (38)	0.79
TP53 mutation, n (%) N = 65	34 (60)	2 (25)	0.01

IQR = interquartile range. MMR = mismatch repair.

## Data Availability

The original contributions presented in this study are included in the article. Further inquiries can be directed to the corresponding author.

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
