# Peer review of "Clinical and Molecular Profiling of Colorectal Cancer: A Comprehensive Cohort Study of BRAF-Mutated Cases from a Tertiary Centre"

_curroncol, 2025, doi:10.3390/curroncol32090507_

Round 1
Reviewer 1 Report
Comments and Suggestions for Authors
Freckelton et al. present a single-centre retrospective cohort of BRAF-mutated colorectal cancer (CRC) treated between 2014–2022 (n=406; 253 localized, 153 metastatic). They compare clinicopathologic features and outcomes between V600 vs non-V600 cases as well as those with concurrent RNF43 mutations. They find that patients with V600 disease were older, right-sided, and more often Mismatch repair (MMR) deficient while patients with non-V600 showed longer time-to-relapse (p=0.006). In metastatic setting, non-V600 cases showed longer OS (34 vs 14 months). Among 158 patients tested on NGS panel including RNF43, 30% had RNF43 mutations; these were associated with older age, V600 mutations, right-sidedness, and MMR-deficiency. On encorafenib+cetuximab, RNF43-mutant cases had higher (100% vs 54%) disease control response (DCR). Authors conclude BRAF mutation class and RNF43 status help explain heterogeneity in BRAF-mutant CRC and suggest routine RNF43 testing for personalized therapy.
Major suggestions:
- Clarify why 37 cases with loss to follow-up within 12 months were excluded rather than censored – authors should justify the potential bias from exclusion vs censoring.
- Methods lack details on NGS profiling, please provide full details of NGS profiling and the bioinformatics pipeline (panel/platform, calling/filters, variant allele frequency thresholds), how variants were categorized (define loss-of-function), and how mismatch repair (MMR) was assessed.
- Provide details on response assessment including RECIST version, imaging schedule, reader blinding (≥2 readers?), the exact DCR definition and evaluation window, and how missing data were handled?
- For the statistical analyses, report effect sizes with 95% confidence intervals (CI) and adjust for confounders. Include multivariable Cox models with hazard ratios for survival and appropriate adjusted models for DCR, controlling for age, sidedness, therapy, mutation class, etc.
- Add median values with 95% CI on Kaplan-Meier curves, include risk tables and confirm text values match the figures (e.g., 14 vs 34 months).
- Resolve the inconsistency between text (N=31, p=0.01 in lines 156-158) and Figure 3A (N=35, p=0.02) for DCR on Encorafenib (BRAF V600) + Cetuximab (EGFR). Ensure denominators, definitions, and p-values match across text, figures, and captions.
- Recommendation for routine RNF43 testing should be given with caution as presented findings are based on a small retrospective single-center cohort. Rather emphasize the need for external validation.
- Consider sharing de-identified per-patient data (BRAF class, RNF43, MMR, co-mutations, line of therapy, response, survival) in a Supplement to facilitate meta-analysis and external validation.
Minor suggestions:
- Introduction is well written – however, consider adding clinical-guideline or practice-changing references to situate findings within current standards.
- Standardize: “PIK3A” -> “PIK3CA”, “RNF 43” -> “RNF43”, “Kaplan-Meir” -> “Kaplan–Meier” and “survival cures” -> “survival curves”. Correct typos and harmonize spacing/formatting across text, tables, and captions.
- For each figure/table, state the analysis set (evalable N), endpoint defitions (e.g., DCR window) and whether numbers are per-patient vs per-sample.
Author Response
The authors thank reviewer 1 for their thoughtful and constructive comments. Please find itemised replies below:
Major suggestions:
Comment 1: Clarify why 37 cases with loss to follow-up within 12 months were excluded rather than censored – authors should justify the potential bias from exclusion vs censoring.
Response 1: These patients were excluded due to a lack of availability of any clinically meaningful data.
Comment 2: Methods lack details on NGS profiling, please provide full details of NGS profiling and the bioinformatics pipeline (panel/platform, calling/filters, variant allele frequency thresholds), how variants were categorized (define loss-of-function), and how mismatch repair (MMR) was assessed.
Response 2: Thank you for the suggestion, please find the methods section updated to include this (row 125-133)
Comment 3:Provide details on response assessment including RECIST version, imaging schedule, reader blinding (≥2 readers?), the exact DCR definition and evaluation window, and how missing data were handled?
Response 3: Response assessment was based on the outcomes reported in pre-existing clinical imaging reports from routine care, as this was a retrospective study. No formal RECIST criteria or centralized radiological review were applied. Disease control rate (DCR) was defined as the presence of stable disease or tumour response as documented at the first restaging imaging. The timing of imaging followed standard clinical practice rather than a fixed schedule. (Row 119-121)
Comment 4: For the statistical analyses, report effect sizes with 95% confidence intervals (CI) and adjust for confounders. Include multivariable Cox models with hazard ratios for survival and appropriate adjusted models for DCR, controlling for age, sidedness, therapy, mutation class, etc.
Response 4:
We thank the reviewer for this thoughtful suggestion. We agree that multivariable models and reporting of effect sizes with 95% confidence intervals are important for robust statistical analysis, particularly when evaluating survival outcomes and disease control rates.
Our primary aim in this study was to provide an exploratory overview of survival differences between BRAF V600 and non-V600 mutated colorectal cancer, rather than to develop a comprehensive prognostic model. Accordingly, our analysis focused on descriptive and univariate comparisons to highlight the differences between these biologically distinct subgroups. Multivariable Cox regression and adjusted models were not initially included due to the relatively small sample size, particularly within the non-V600 subgroup, limited our ability to perform reliable multivariable analyses without the risk of overfitting.
We fully acknowledge that adjusting for confounders such as age, tumor sidedness, therapy, and mutation class would provide additional insights, and we have now added this as a limitation in the revised Discussion section. (row 250-251)
Comment 5: Add median values with 95% CI on Kaplan-Meier curves, include risk tables and confirm text values match the figures (e.g., 14 vs 34 months).
Response 5: Thank you this has been addressed.
Comment 6: Resolve the inconsistency between text (N=31, p=0.01 in lines 156-158) and Figure 3A (N=35, p=0.02) for DCR on Encorafenib (BRAF V600) + Cetuximab (EGFR). Ensure denominators, definitions, and p-values match across text, figures, and captions
Response 6: Thank you this has been addressed.
Comment 7: Recommendation for routine RNF43 testing should be given with caution as presented findings are based on a small retrospective single-center cohort. Rather emphasize the need for external validation.
Response 7: Thank you, this recommendation has been qualified in the discussion (row 267-268)
Comment 8: Consider sharing de-identified per-patient data (BRAF class, RNF43, MMR, co-mutations, line of therapy, response, survival) in a Supplement to facilitate meta-analysis and external validation.
Response 8: Thank you for the thoughtful suggestion. We recognise the value that sharing de-identified per-patient data could provide for meta-analyses and external validation. However, the ethics approval under which this study was conducted does not permit the sharing of individual-level patient data.
Minor suggestions:
Comment 1: Introduction is well written – however, consider adding clinical-guideline or practice-changing references to situate findings within current standards.
Response 1: Thank you for the suggestion, introduction has been updated to include new data including BREAKWATER study
Comment 2: Standardize: “PIK3A” -> “PIK3CA”, “RNF 43” -> “RNF43”, “Kaplan-Meir” -> “Kaplan–Meier” and “survival cures” -> “survival curves”. Correct typos and harmonize spacing/formatting across text, tables, and captions.
Response 2: Thank you for finding these errors, they have been addressed
Comment 3: For each figure/table, state the analysis set (evaluable N), endpoint definitions (e.g., DCR window) and whether numbers are per-patient vs per-sample.
Response 3: Thank you addressed
Reviewer 2 Report
Comments and Suggestions for Authors
The authors have performed a retrospective cohort study investigating the clinical and molecular data of 406 patients with BRAF-mutated colorectal cancer. The results reveal significant differences between the characteristics of V600 and non-V600 BRAF mutated tumors, which are in line with published literature. The demonstrated association between co-occurring BRAF/RNF43 mutations and higher disease control rate with combined BRAF/EGFR targeted therapy has been previously published, but publication of the current report is important to validate this clinically important finding in an independent study cohort.
Minor points:
-Abstract: It would help the reader, if the authors state already in the Methods part that V600 mutations were compared to non-V600 mutations. Row 28, 20% in metastatic.
-Rows 26 and 134: PIK3CA.
-Rows 49-50: Gene/protein names other than RAS are not spelled out. The “virus” is a bit confusing here and I suggest leaving the long form out.
-Rows 52-53: Please add a reference.
-Rows 64-70: Please add a reference.
-Rows 71-73: Please add a reference. CpG islands.
-Row 74: Promoter hypermethylation leads to silencing of gene expression.
-Table 1: Please spell our abbreviations below the table. The total number of V600BRAFmt dMMR cases is 127, but the sum of the subclasses (117+7+2+2) is 128. Please correct. Adjuvant systemic treatment, V600BRAFmt 100/228 should be 44%.
-Table 2: It’s not very clear how the percentages of KRAS/NRAS/PIK3CA/TP53/RNF43 mutation part have been calculated. Are the N-values total number of analyzed samples? Please make this part easier for the reader, e.g. using a footnote. Please check the percentages (e.g. non-V600BRAFmt males, dMMR classes). Also here, dMMR n=28, but the sum of subclasses is 27.
-Figure 2: Here the mutation classes are typical and atypical, everywhere else V600 and non-V600.
-Table 3: Please check the percentages. Explain what are the N-values related to KRAS/PIK3CA/TP53.
-Row 153: TP53 mutations.
-Rows 156-160: Please cite Fig. 3.
-Figure 3: In the legend of 3A, n=35, whereas in the text 31 is mentioned. How many of these were BRAFmt and BRAFwt? How about patients in 3B?
-Figure 4: Mentioning "metastatic cases" in the legend?
-Row 188: 20% in metastatic, 56% in localized.
-References: Some references are quite old. Please update with more recent ones (e.g. reviews 3 and 12).
Author Response
The authors thank reviewer 2 for the constructive comments. Please see responses below
Comment 1: It would help the reader, if the authors state already in the Methods part that V600 mutations were compared to non-V600 mutations. Row 28, 20% in metastatic.
Response 1: Thank you, this has been addressed in row 32
Comment 2: Rows 26 and 134: PIK3CA.
Response 2: Thank you, terminology discrepancy has been addressed
Comment 3: Gene/protein names other than RAS are not spelled out. The “virus” is a bit confusing here and I suggest leaving the long form out.
Response 3: This has been removed as suggested
Comments 4-6: Please add references
Response 4-6: Thank you, references have been added
Comments 7: Promoter hypermethylation leads to silencing of gene expression.
Response 7: This has been addresed in the text (row 93)
Comment 8: Table 1: Please spell our abbreviations below the table. The total number of V600BRAFmt dMMR cases is 127, but the sum of the subclasses (117+7+2+2) is 128. Please correct. Adjuvant systemic treatment, V600BRAFmt 100/228 should be 44%.
Response 8: Thank you this has been addressed in the table
Comment 9: Table 2: It’s not very clear how the percentages of KRAS/NRAS/PIK3CA/TP53/RNF43 mutation part have been calculated. Are the N-values total number of analyzed samples? Please make this part easier for the reader, e.g. using a footnote. Please check the percentages (e.g. non-V600BRAFmt males, dMMR classes). Also here, dMMR n=28, but the sum of subclasses is 27.
Response 9: Thank you for the feedback, the comments have been addressed in table 2
Comment 10: Figure 2: Here the mutation classes are typical and atypical, everywhere else V600 and non-V600.
Response 10: Thank you this has been addressed for consistency (figure 2)
Comment 11: Table 3: Please check the percentages. Explain what are the N-values related to KRAS/PIK3CA/TP53.
Response 11: Thank you, this has been adressed in the table legend
Comment 12: Row 153: TP53 mutations.
Response 12: Amended
Comment 13: Please cite Fig. 3.
Response 13: Fig 3A + 3B cited
Comment 14: Figure 3: In the legend of 3A, n=35, whereas in the text 31 is mentioned. How many of these were BRAFmt and BRAFwt? How about patients in 3B?
Response 14: Thank you this has been addressed in the text
Comment 15: Figure 4: Mentioning "metastatic cases" in the legend
Response 15: Legend clarified
Comment 16: 20% in metastatic, 56% in localized.
Response 16: Thank you addressed
Comment 17: References: Some references are quite old. Please update with more recent ones
Response17: Updated
Round 2
Reviewer 1 Report
Comments and Suggestions for Authors
I thank authors for making necessary changes and revising their manuscript.
please note that comment 6 has not been properly addressed, please resolving inconsistencies between p-values in text (p=0.01) and Figure 3A (p=0.02) for DCR on Encorafenib (BRAF V600) + Cetuximab (EGFR).
Author Response
Comment 1: please note that comment 6 has not been properly addressed, please resolving inconsistencies between p-values in text (p=0.01) and Figure 3A (p=0.02) for DCR on Encorafenib (BRAF V600) + Cetuximab (EGFR).
Response 1: Thank you very much for identifying this error, it has now been addressed.